# Consensus based framework for digital mobility monitoring

**Felix Kluge**[1]*, **Silvia Del Din**[2], **Andrea Cereatti**[3], **Heiko Gaßner**[4], **Clint Hansen**[5], **Jorunn L. Helbostad**[6], **Jochen Klucken**[4], **Arne Küderle**[1], **Arne Müller**[7], **Lynn Rochester**[2,8], **Martin Ullrich**[1], **Bjoern M. Eskofier**[1], **Claudia Mazzà**[9], on behalf of the Mobilise-D consortium[¶]

**1** Machine Learning and Data Analytics Lab, Department Artificial Intelligence in Biomedical Engineering, Friedrich-Alexander University Erlangen-Nürnberg, Erlangen, Germany, **2** Translational and Clinical Research Institute, Faculty of Medical Sciences, Newcastle University, Newcastle upon Tyne, United Kingdom, **3** Department of Electronics and Telecommunications, Politecnico di Torino, Torino, Italy, **4** Department of Molecular Neurology, University Hospital Erlangen, Erlangen, Germany, **5** Department of Neurology, University of Kiel, Kiel, Germany, **6** Department of Neuromedicine and Movement Science, Faculty of Medicine and Health Sciences, Norwegian University of Science and Technology, Trondheim, Norway, **7** Novartis Pharma AG, Basel, Switzerland, **8** The Newcastle upon Tyne NHS Foundation Trust, Newcastle upon Tyne, United Kingdom, **9** Department of Mechanical Engineering & Insigneo Institute for in Silico Medicine, The University of Sheffield, Sheffield, United Kingdom

¶ Membership of the Mobilise-D consortium is provided in the Acknowledgments.
* felix.kluge@fau.de

**Data Availability Statement:** The datasets are available on the Zenodo database (https://doi.org/10.5281/zenodo.4316564).

**Funding:** All authors are part of the MOBILISE-D project. The MOBILISE-D project has received

## Abstract

Digital mobility assessment using wearable sensor systems has the potential to capture walking performance in a patient's natural environment. It enables monitoring of health status and disease progression and evaluation of interventions in real-world situations. In contrast to laboratory settings, real-world walking occurs in non-conventional environments and under unconstrained and uncontrolled conditions. Despite the general understanding, there is a lack of agreed definitions about what constitutes real-world walking, impeding the comparison and interpretation of the acquired data across systems and studies. The goal of this study was to obtain expert-based consensus on specific aspects of real-world walking and to provide respective definitions in a common terminological framework. An adapted Delphi method was used to obtain agreed definitions related to real-world walking. In an online survey, 162 participants from a panel of academic, clinical and industrial experts with experience in the field of gait analysis were asked for agreement on previously specified definitions. Descriptive statistics was used to evaluate whether consent (> 75% agreement as defined a priori) was reached. Of 162 experts invited to participate, 51 completed all rounds (31.5% response rate). We obtained consensus on all definitions ("Walking" > 90%, "Purposeful" > 75%, "Real-world" > 90%, "Walking bout" > 80%, "Walking speed" > 75%, "Turning" > 90% agreement) after two rounds. The identification of a consented set of real-world walking definitions has important implications for the development of assessment and analysis protocols, as well as for the reporting and comparison of digital mobility outcomes across studies and systems. The definitions will serve as a common framework for

funding from the Innovative Medicines Initiative 2
Joint Undertaking under grant agreement No.
820820. This Joint Undertaking receives support
from the European Union's Horizon 2020 research
and innovation program and the European
Federation of Pharmaceutical Industries and
Associations (EFPIA). SDD and LR are supported
by the National Institute for Health Research
(NIHR) Newcastle Biomedical Research Centre
(BRC) based at Newcastle Upon Tyne Hospital
NHS Foundation Trust and Newcastle University.
The work was also supported by the NIHR/
Wellcome Trust Clinical Research Facility (CRF)
infrastructure at Newcastle upon Tyne Hospitals
NHS Foundation Trust. AM is employed by the
Novartis Pharma AG. The funder provided support
in the form of salary for AM, but did not have any
additional role in the study design, data collection
and analysis, decision to publish, or preparation of
the manuscript. BME gratefully acknowledges the
support of the German Research Foundation (DFG)
within the framework of the Heisenberg
professorship program (grant number ES 434/8-
1). CM is supported by the National Institute for
Health Research (NIHR) Sheffield Biomedical
Research Centre (BRC) based at Sheffield Teaching
Hospital NHS Foundation Trust and University of
Sheffield. All opinions are those of the authors and
not the funders. The funders had no role in study
design, data collection and analysis, decision to
publish, or preparation of the manuscript. The
specific roles of the authors are articulated in the
'author contributions' section.

**Competing interests:** The authors have declared
that no competing interests exist. The funders did
not have any role in the study design, data
collection and analysis, decision to publish, or
preparation of the manuscript. The commercial
affiliation does not alter our adherence to PLOS
ONE policies on sharing data and materials.

implementing digital and mobile technologies for gait assessment and are an important link for the transition from supervised to unsupervised gait assessment.

## Introduction

Mobility, or specifically gait, can be influenced by a variety of chronic health conditions, spanning from neurological, respiratory, and cardiac to musculoskeletal disorders. Such conditions may include multiple sclerosis (MS), Parkinson's disease (PD), chronic obstructive pulmonary disease (COPD), congestive heart failure (CHF), or proximal femoral fracture (PFF) [1–6]. Related functional mobility impairments present a great burden to patients, severely limiting quality of life [7–9], alongside an increased fall risk [10–12], and mortality [13, 14].

Changes in various gait measures such as cadence, gait speed, and stride length amongst others may characterize those mobility impairments. The use of digital mobility outcomes (DMOs), which we refer to as digital measures acquired using digital health technology [15] has already been studied in clinical settings using brief, standardized tests in a range of diseases [2, 4, 16, 17]. However, a single observation may not be reliable for clinical characterization especially when mobility related disease symptoms fluctuate over acute periods of time. Therefore, the objective assessment of gait calls for valid and reliable methods to sensitively capture changes in gait function more frequently [18]. As it is not feasible to increase patient visits to the clinic, more continuous monitoring outside laboratory or clinical environments is desired [19]. Thus, the continuous assessment of real-world digital measures is essential and opens the opportunity for frequent and long-term remote monitoring [18, 20–22]. In the past years, real-world gait analysis has been technologically enabled by the development of lightweight and easy to use sensor-based systems that can be worn unobtrusively. Although DMOs quantified from real-world data are able to discriminate and detect gait impairments in various diseases [20, 23–26], accepted and routinely used tools are not applied in practice yet [27].

Whilst real-world measurement of mobility holds promise, one fundamental reason for the lack of adoption is the difficulty of comparing DMOs across studies due to the inconsistent use of terminology. As an example, a broad variety of terms describing the real-world context exist, including real-life, daily-life, everyday-life, and free-living [19, 20, 28, 29]. These terms are used interchangeably with ambiguous definitions, leading to different test paradigms being considered and impeding comparability across measurements, systems, and studies. Furthermore, observed DMO variations may not only be caused by disease symptoms but also environmental factors and measurement protocols, which affect the reliability of DMO assessment. Therefore, agreed definitions of relevant DMOs and the context of measurement are necessary to guarantee clinical meaningfulness. As a further example, the term walking bout has been used in the context of real-world gait analysis and refers to the quantification of continuous periods of free-living walking [30]. However, walking bout definitions are inconsistent and may include different walking bout durations and number of strides [10, 12, 28, 30–32]. The duration of resting periods between walking bouts [33], and whether turning is considered as part of walking [34, 35] are treated differently as well. However, a clear definition of a walking bout is critical, since it directly affects digital measures [28, 36]. Additionally, turning needs to be considered as a main constituent of walking, as an average of more than 60 turns per hour has been reported for real-world walking [34]. Due to their high occurence, turnings are likely to break sequences of straight walking into smaller walking bouts. Therefore, the specific definition of a turn directly influences the distribution of walking bouts with regard to their duration. Furthermore, spatio-temporal parameters during straight walking and turning differ [37], such that real-world DMOs based on averages of those parameters

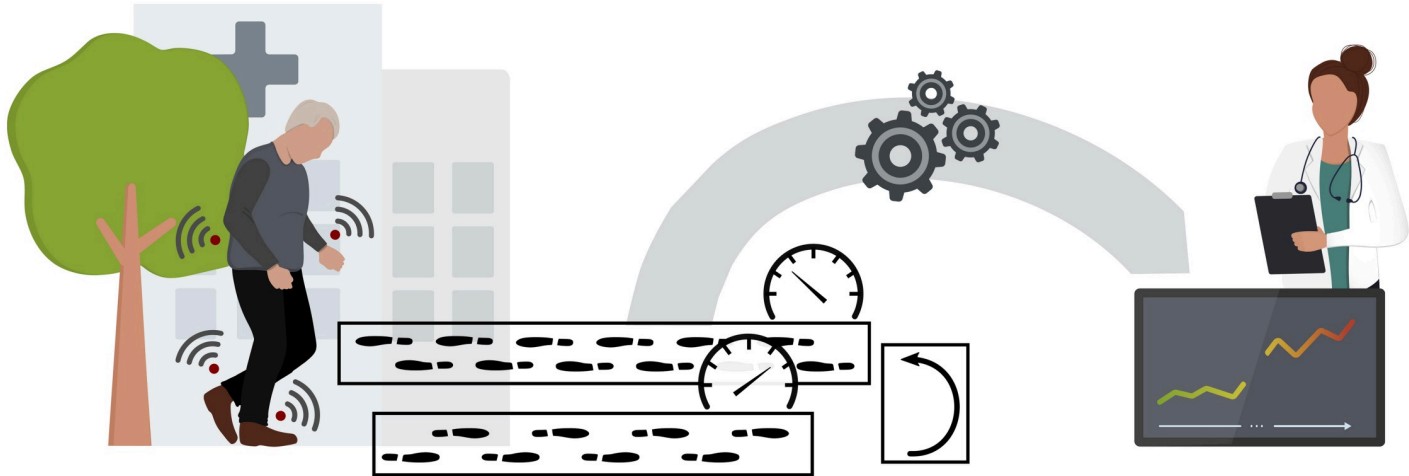

**Fig 1. Real-world walking assessment.** Monitoring of real-world mobility using wearable sensors requires the definition of essential components of unsupervised purposeful walking, such as walking bouts and turnings. The aggregated walking bouts at different speed levels yield digital mobility outcomes which can characterize clinically relevant changes in mobility impairments.

strongly depend on whether turning is included in their estimation. Currently, different operational approaches exist for the detection of turning. As an example, turning characteristics may be based on stride to stride angular parameters using foot rotation [38] or on angular changes related to the trunk rotation [34]. This diversity highlights the need for defining turning in a framework of DMO assessment, which will anable a more specific operationalization of real-world DMOs.

While the above terms are ambiguously used in previous research, there is no integrative taxonomy regarding real-world walking yet. Thus, there lacks a guiding framework that can be used for further implementation of DMOs in real-world settings. Accordingly, the aim of this study was to build a terminological framework in order to drive the development and assessment of DMOs for real-world monitoring (Fig 1). We aimed to reach agreement upon a set of narrative definitions within the scope of the Mobilise-D project [15], which is a five-year EU-funded IMI consortium that will build a technically and clinically-valid system for real-world digital mobility assessment across multiple populations with the goal to improve healthcare.

In this study, we used an objective and systematic consensus process based on an adapted Delphi method [39]. Our results will enable operational definitions to implement mobility assessment algorithms, foster comparability across studies, and serve as a common communication framework for the scientific community. Perspectively, consensus on such a terminological framework is a prerequisite for the adoption of validated digital biomarkers characterizing mobility impairments in various diseases [15, 27].

## Materials and methods

Our approach of defining a terminological framework consisted of the following steps: First, relevant domains and key terms related to real-world walking for the consensus process were identified. Six terms related to four domains of real-world walking needing group consensus were selected (Table 1). For some terms, different aspects were regarded. We proposed a physiological definition of walking and highlighted its relationship to walking bouts. For the definition of real-world, we defined fundamental characteristics, how it is discriminated from standardized measurements and which test paradigms in the clinical context may be regarded

**Table 1. Identified domains and terms needing consensus.**

| Domain | Term | Aspect |
|---|---|---|
| *What* are you doing? | Walking | Physiological |
| | | Relation to walking bouts |
| *Why* are you doing it? | Purposeful | Characteristics |
| *Where* are you doing it? | Real-world | Characteristics |
| | | Clinical environment |
| | | Standardized measurement |
| *How* are you doing it? | Walking bout | Characteristics |
| | Walking speed | Physical definition |
| | | Granularity |
| | | Relation to walking bouts |
| | Turning | Characteristics |

as real-world assessment. The walking speed definition was based on physical considerations. Additionally, we identified the need to consider different granularities when calculating aggregated speed measures and proposed that real-world walking speed needs to be inherently connected to walking bouts. We proposed initial definitions for those eleven aspects based on the study team's expert knowledge and literature. Iterative feedback was included to improve structure and content of the definitions (for the questionnaire with the initial definitions, see S1 File). We used these definitions as starting point for the subsequent consensus process.

We adopted an objective consensus building process based on the Delphi methodology [39]. In contrast to quantitative methods such as systematic reviews or meta-analyses, which are based on available literature and studies, the Delphi process allows to obtain consensus among experts by determining the level of agreement on a given topic [39]. Specifically, the Delphi method is characterized by anonymity to avoid dominance of single experts, multiple iterations, and feedback to the group. As such, a basic Delphi technique can contain any type of self-administered questionnaire with no meetings [40], which is the approach that was used in this explorative study to quantitatively assess the agreement on the initial definitions.

A consensus process may consist of multiple rounds until agreement on the definitions is reached. Based on a 5-point Likert scale (see S1 File for the initial definitions including the used scale), agreement was quantitatively assessed among the participants [41]. A neutral statement ("No opinion") was included. In our study, agreement to a given definition was defined a-priori as more than 75% of the answers belonging to the categories "Somewhat agree" or "Strongly agree" [40–42]. In the first round, the participants were asked to independently rate all eleven statements across the six key terms "Walking", "Purposeful", "Real-world", "Walking bout", "Walking speed", and "Turning". Additionally, participants were asked to provide free-text comments for each item in order to capture input for the improvement of definitions [41]. In subsequent rounds, we presented modified definitions that previously did not reach agreement and reassessed the agreement.

The consensus process was performed among members of the Mobilise-D consortium. The project includes technical, clinical, and industrial expertise of 34 partners from Europe and the USA. All 162 members of the consortium were asked for participation via email. Participants who did not respond in the first round were not invited to participate in the second round. We did not define any exclusion criteria. Data on participants' technical or clinical background, gait expertise, free-living expertise, and expertise with patients were collected in the first round to analyse the panel's background.

The consensus process was implemented as a series of questionnaires based on the "Survey" feature of the ILIAS e-Learning platform (version 5.4.5, ILIAS open source e-Learning e.V.). It allowed the anonymous acquisition of responses. The participants' email addresses were linked to access codes, which were provided to start the questionnaire. The use of access codes allowed sending reminders to the participants and preventing double participation. The acquisition and analysis of the data was anonymous. Descriptive statistics was used to investigate participants' background information and agreement responses in each round.

Analyses were conducted using R version 4.0.3 [43]. The code is available on https://doi.org/10.5281/zenodo.4316739. The datasets generated and analysed in the study are available on https://doi.org/10.5281/zenodo.4316564.

Ethical approval for this study was granted by the ethics committee of the University Hospital Erlangen (Re.-No. 241_19 Bc). All participants provided written informed consent before inclusion in the study. Participation in this study was voluntary. All data were handled in accordance with European data protection regulations.

## Results

### Consensus process

In total, the consensus process required two rounds until agreement on all definitions was reached. 162 members of the Mobilise-D consortium were asked to participate in the first round of the consensus process. Of those, 79 individuals started the questionnaire. Eight individuals did not sign the participation or data usage agreement and five participants did not complete the questionnaire. Hence, their data was discarded. Data from the remaining 66 participants (40.7% response rate in the first round) were analysed. Of the participants who completed the first round, 55 individuals started the second round. One individual did not sign the participation agreement and three individuals did not complete the questionnaire. We analysed the data of the remaining 51 participants (continuation response rate of 77.3%). The overall response rate was 31.5%. The professional background of the panel was diverse but homogenously distributed across clinical and technical disciplines (Table 2). Only 12.1 stated to have no experience in gait analysis. Two thirds of all individuals had expertise in real-world mobility. Most participants (88.7%) stated to have expertise with patients. As answering the background questions was not obligatory, the total number differs from the total number of individuals who participated in round one of the process.

### Agreement on definitions

In round one, the definitions of purposeful and walking speed (relation to walking bouts) did not reach agreement (Table 3) and were subject to modification based on participants' feedback. Although the definition of a walking bout reached agreement, there was inclarity regarding its inherent connection to the walking speed definition. The walking speed definition initially assumed a different number of strides required to assess average walking speed (for the initial definitions, see S1 File). Therefore, we decided to harmonize the walking bout and walking speed (relation to walking bouts) definitions, which were both put to vote again in the second round. Full consensus for all definitions was reached in round two (Table 3) resulting in a final set of definitions for real-world gait analysis (Table 4).

## Discussion

To the best of the authors' knowledge, this has been the first study to engage clinicians, academic researchers, and industry stakeholders working in the field of digital gait and mobility

**Table 2. Participant background assessed in the first round.**

| | |
|---|---:|
| **Professional background (n = 64)** | |
| Both technical and clinical background | 25.0% |
| Clinical background | 40.6% |
| Technical background | 34.4% |
| **Expertise in gait analysis (n = 66)** | |
| None | 12.1% |
| 0–5 years | 50.0% |
| 5–10 years | 12.1% |
| 10–15 years | 7.6% |
| 15–20 years | 4.6% |
| > 20 years | 13.6% |
| **Free-living expertise (n = 65)** | |
| No | 35.4% |
| Yes | 64.6% |
| **Clinical expertise (n = 62)** | |
| Patients and healthy participants | 71.0% |
| Healthy participants | 11.3% |
| Patients | 17.7% |

measures development to identify and agree upon a framework of narrative definitions for the assessment of DMOs acquired in real-world conditions. An adapted Delphi consensus process allowed to achieve consensus on eleven statements related to six key terms of real-world walking.

A broad definition of walking may include various displacements of the body in space (e.g., for-, back-, or sideward walking). However, we defined walking to be only associated with forward displacement using both legs in order to assure reliability of DMO assessment in various contexts. Stepping on the spot, side stepping, and backward walking have thus been excluded from this definition. Walking is also not defined in terms of a specific speed. The use of walking aids has been included into the definition as they may be an essential requirement for safe locomotion of people with gait impairments. Otherwise, certain patients and elderly individuals might be excluded from the DMO assessment. We acknowledged steps and strides as basic elements of walking as previously suggested [49]. Furthermore, the definitions include that walking is always made up of walking bouts, where it was agreed that walking bouts represent sequences containing at least two full consecutive strides of both feet without a break (e.g., *R-L-R-L-R-L* or *L-R-L-R-L-R*, with *R/L* being the contacts of the right/left foot with the ground, respectively). The start and end of a walking bout are determined by a break that can either consist of a resting period, turning, or any other non-walking real-world activity. More specifically, the start is always defined by an initial step of a walking bout following a non-walking period, while the final step precedes the next non-walking period. Walking bouts are thus an important building block in the terminology framework for the assessment of DMOs acquired in real-world conditions. Furthermore, this definition can equally be used in the context of supervised clinical and functional assessment, which currently is the clinical reference for mobility assessment.

Walking speed has been referred to as sixth vital sign, as a slower walking speed has been associated with morbidity, cognitive decline, and fall risk amongst others [13, 50]. Despite this, there is still no accepted common measure of mobility that serves across multiple conditions,

**Table 3. Proportion of agreement and disagreement [%] of definitions.** In round two, only those definitions were evaluated, which did not reach agreement in the first round. The lower limit of agreement was a priori defined as 75%.

| Term | Aspect | Disagreement | No opinion | Agreement | Consensus |
|------|--------|-------------|-----------|----------|-----------|
| | | **Round one** | | | |
| Walking | Physiological | 6.1 | 3.0 | 90.9 | yes |
| | Relation to walking bouts | 0.0 | 3.0 | 97.0 | yes |
| Purposeful | Characteristics | 10.6 | 21.2 | 68.2 | no |
| Real-world | Characteristics | 4.6 | 1.5 | 93.9 | yes |
| | Clinical environment | 4.6 | 1.5 | 93.9 | yes |
| | Standardized measurement | 4.6 | 3.0 | 92.4 | yes |
| Walking bout | Characteristics | 15.1 | 6.1 | 78.8 | yes |
| Walking speed | Physical definition | 1.5 | 1.5 | 97.0 | yes |
| | Granularity | 1.5 | 6.1 | 92.4 | yes |
| | Relation to walking bouts | 15.2 | 24.2 | 60.6 | no |
| Turning | Characteristics | 3.0 | 3.0 | 94.0 | yes |
| | | **Round two** | | | |
| Purposeful | Characteristics | 15.7 | 5.9 | 78.4 | yes |
| Walking bout | Characteristics | 11.8 | 5.9 | 82.3 | yes |
| Walking speed | Relation to walking bouts | 15.7 | 7.8 | 76.5 | yes |

which is underlined by a wide range of inconsistent testing procedures. With an operative definition of walking speed and a proposition of respective aggregation levels at which it is measured, we aim to provide a common framework to be used across clinical conditions. The physical definition of walking speed reached high consensus, where the panel also agreed that walking speed will be assessed based on a minimal number of consecutive strides. According to clinical questions, walking speed needs to be assessed with regard to different aggregation levels (hourly, daily, weekly, etc.). This definition is in line with the walking bout definition. On the one hand, this specification yields a unified approach of assessing speed in our framework, but requires a stride-wise analysis of walking, which might not be feasible in all analysis cases, for example when the extraction of single strides is not possible. The definition also suggests that walking speed derived from strides that are not part of walking bouts (i.e., short strides, shuffling, turning, etc.) should not be considered for the estimation of real-world walking speed.

Daily mobility does not only contain straight walking but also curved walking and turns. Therefore, we included a definition of turning in the framework to guide the implementation of walking bouts and the related DMO assessment. Turning can be regarded as being a deceleration of the forward motion, rotating the body as a whole, and stepping out toward the new direction [48]. It results in a change of walking direction and change in angular orientation including a rotational movement of the body around the longitudinal axis. As an example, a threshold on the rotation angle (e.g., > 45˚) at a certain turn duration (e.g., between 0.5s and 10s) can be used to detect a turn [34]. This definition does not include all the required aspects for quantitative ambulatory mobility measurement. For example, further discussion will be necessary to define specific angular thresholds between straight and non-straight (i.e., curvilinear) walking. However, those operational definitions are not part of the narrative framework considered in this study and will be evaluated based on real-world data from different clinical populations in future work.

The environmental context of walking greatly influences DMOs. Thus, it was deemed necessary to specify inclusion and exclusion criteria of what is considered real-world. The

**Table 4. Agreed definitions of terms related to real-world walking.**

| Term | Aspect | Definition |
|---|---|---|
| **Walking** | Physiological | Human **walking** is a method of locomotion and is defined as initiating and maintaining a forward displacement of the centre of mass in an intended direction involving the use of the two legs, which provide both support and propulsion. The feet are repetitively and reciprocally lifted and set down whereby at least one foot is in contact with the ground at all times [44, 45]. **Walking** with walking aids is included in this definition.<br>A **step** is the interval between the initial contacts of the ipsi- and contralateral foot [44] and corresponds to the forward displacement of the foot together with a forward displacement of the trunk [46].<br>A **stride** is the interval between two successive initial contacts of the same foot. As such, a **stride** is equivalent to the gait cycle and every stride contains two **steps** [44]. |
| | Relation to walking bouts | **Walking** is made up of walking bouts and is equivalent to taking steps/stepping forward (thus stepping in place does not constitute walking) and is defined as starting from initial contact for the initial step until ending with full floor contact of the foot making the last step [46]. |
| **Purposeful** | Characteristics | **Purposeful walking** includes an intentional component of the movement (e.g., getting to the bathroom, catching the bus, going to the grocery store, going for a walk in the park, etc.).<br>**Purposeful walking** may constitute certain characteristics (e.g., more constant walking velocity, lower variability of gait characteristics, straighter direction of locomotion than non-purposeful walking, specific context, etc.). Those gait characteristics are quantified based on discrete walking bouts. |
| **Real-world** | Characteristics | **Real-world** relates to the context in which walking takes place—that is free-living, unsupervised, uncontrolled and non-standardised. As such, it is unscripted as there are no instructions to the subject who does not need to interact with the wearable device(s).<br>**Real-world** actions occur in non-simulated everyday situations in unconstrained environments with minimal consciousness of being tested. It is equivalent to actions at home or in the community over continuous periods of time [28].<br>Synonymous terms are (environment of) **daily living**, or relating to **daily-life**. **Home environment** is used synonymously to **real-world** and **daily-life** without a separation of indoor and outdoor environment [11].<br>**Real-world** is distinct from laboratory-based [47], supervised (= fully controlled and observed), and semi-controlled (walking 'freely' but with supervision) tests. It also is different from scripted/instructed walking, which can take place in the home or lab (such as walking tests like the 4x10m test, 6-minute walk test (6MWT) and timed up and go (TUG)). |
| | Clinical environment | Free walking in hospitals is part of the **real-world** definition, but standardized supervised tests in a hospital are not. This excludes instructed actions, e.g., by medical professionals. |
| | Standardized measurement | Home-based tests, which are semi-standardized measurements performed in the home environment in a controlled or semi-controlled environment (such as short walk tests), are thus not regarded as being part of **real-world**. Home-based tests can nevertheless be an alternative to clinical tests and might be easier to conduct operationally and analyse than continuous monitoring (assuming standardized instructions). |
| **Walking bout** | Characteristics | A **walking bout (WB)** is a walking sequence containing at least two consecutive strides of both feet (e.g., *R-L-R-L-R-L* or *L-R-L-R-L-R*).<br>Start and end of a **walking bout** are determined by a resting period or any other activity (non-walking period). The initial step of a **WB** follows a non-walking period and the final step precedes the next non-walking period. |
| **Walking speed** | Physical definition | **Walking speed (WS)** is the distance covered by the whole body within a certain time interval / per unit time of walking. It is measured in meters per second and is the magnitude of the velocity vector (velocity includes direction and magnitude of walking) [45]. |
| | Granularity | **Walking speed** can be estimated at different granularities:<br>• Instantaneous WS varies from one instant to another during the walking cycle [45]<br>• Step-wise WS is the ratio between step distance (length) and step time [28]<br>• Stride-wise WS [33]<br>• Averaged over WBs<br>• Averaged over other time intervals (hourly, daily, weekly) based on multiple WBs<br>The granularity by which the WS is assessed should be related to clinical parameters for each population separately. |
| | Relation to walking bouts | **Walking speed** will be assessed with regard to walking bouts. Thus, the minimal length of one walking bout required to assess average walking speed is based on a sequence of 2 consecutive strides (e.g., *R-L-R-L-R-L* or *L-R-L-R-L-R*). |
| **Turning** | Characteristics | The process of **turning** consists of decelerating the forward motion, rotating the body as a whole, and stepping out toward the new direction [48]. Thus, **turning** includes a change of walking direction and change in angular orientation including a rotational movement of the body around the longitudinal axis. **Turning**, curvilinear walking, and straight walking involve different neuromotor strategies and need to be discriminated. |

participants agreed on different aspects of the terminology: real-world is conceptualized as free-living, unsupervised, uncontrolled, and non-standardized. In the real-world context, the measurement of DMOs should not interfere with daily activities of the participant. The measurement process of real-world DMOs should thus be as non-obtrusive as possible. Accordingly, this definition is distinct from laboratory-based [47], supervised (fully controlled and observed), and semi-controlled (walking freely but with supervision) environments, in which observer and instruction effects might occur and influence DMOs. For example, walking happens in non-simulated real-world situations in unconstrained environments equivalent to actions at home or in the community over continuous periods of time [28]. Daily-living, including the home and clinical environment are equivalent to real-world as long as the walking happens unsupervised. Scripted walking capacity tests such as 4x10 m walking conducted at home are excluded from the definition, as significant differences between DMOs derived from those tests and DMOs acquired during unscripted real-world walking are expected [51]. However, relationships between standardized tests and real-world assessments still need to be evaluated in future research studies.

The participants agreed to the definition of purposeful as a consistent term for the assessment of DMOs acquired in real-world conditions. Purposeful walking includes an intentional component of the movement. We assume that unsupervised walking is per se purposeful and that the intentional aspect occurs especially for long walking bouts and needs to be evaluated taking for example contextual aspects of walking into account. Differences between purposeful, self-initiated movements, and movements performed in a supervised (and thus not real-world setting) are discussed in detail in [52], and should be further investigated with the consented real-world walking definitions.

The definitions agreed upon in this study build a framework in order to capture gait analysis characteristics and properties in the real-world environment. The goal to have working definitions for various clinical populations has resulted in a rather broad definition of walking (e.g., inclusion of walking aids). However, clarity on specific parts of the definition (e.g., that walking only includes forward locomotion) will allow to implement very specific digital mobility measures without restricting the application cases.

While the Delphi approach is commonly used to obtain broad consensus among experts by determining the level of agreement on a given topic [39], there is always a certain bias. We used purposive sampling under the assumption that members of the Mobilise-D consortium represented experts in the field of real-world gait research. Although we acknowledge that the choice of participants limits generalizability of the results, the consortium includes a large group of experts on gait analysis from Europe and the USA. The participants were homogeneously distributed regarding technical and clinical background, where their views were gathered from a wide range of clinical and academic disciplines to equally represent a breadth of expertise. Some participants stated no experience with gait analysis before. However, most of the participants explicitly mentioned having worked in the field of real-world gait analysis. Moreover, the larger proportion of the participants already had experience in clinical gait analysis. Nevertheless, further work is needed to validate the results of our study in light of an even broader international group of gait experts. For example, the survey could be opened to a wider panel to validate and refine the findings. Furthermore, the framework needs to be evaluated with regard to clinical interpretability of the acquired DMOs based on actual real-world data. Overall, given the geographical spread, the Delphi consensus method conducted online was an appropriate tool for gathering the different viewpoints as compared to physical discussion rounds. One challenge in group decision making is finding an optimal consensus threshold. In our study, we used an a-priori threshold of 75 for assessing the agreement to a given definition which is similar to thresholds previously used [40–42]. Furthermore, we evaluated

consensus based on the proportion of agreement within a range (more than 75 of the answers belonging to the categories "Somewhat agree" or "Strongly agree"). Other definitions of consensus exist and may be taken into account in future studies [42].

The low response rates observed in this study, especially in the first round, are typical for consensus processes and have previously been reported [41]. Especially with large sample sizes, low response rates are considered to be a drawback [53]. Specifically to this study, we invited all members of the Mobilise-D consortium to take part in the study, if they could comment on the topic. Some of the participants invited might not have had a real-world gait analysis background or interest and did therefore not participate in the process. As discussed, the analysis of the participants' professional background showed that the included participants had relevant experience in the field of interest. Furthermore, the final sample size of 51 participants was higher than the lower threshold of 12, which has been regarded as minimal number to ensure reliability of results in a consensus process [54].

Only group feedback in the questionnaires were provided, as individual feedback was not possible due to anonymity. However, the participants were sent an email with their individual responses and comments after completion of a round. This allowed them to reflect on their own ratings in the subsequent round.

One limitation of this study is that the definitions are only of narrative nature. While the obtained definitions have been objectively derived, some may need refinement according to the practical needs to directly guide algorithm implementation (e.g., thresholds on differentiating turning from curvilinear or straight walking need to be derived from further consensus or based on real-world data). Whilst extracting and analysing DMOs, more detailed definitions need to be derived from the initial framework to enlarge the scope and ensure applicability across different technologies and solutions for real-world gait assessment.

This work was conducted as part of the Mobilise-D project [15] with the aim to guide the data analysis process regarding real-world walking analysis with a focus on the assessment of real-world walking speed. It has to be noted, that different ways of assessing real-world mobility exist, such as analyzing daily activity patterns (e.g., daily step count, physical activity, energy expenditure amongst others). Related digital measures are of high interest for some diseases and might benefit from similar terminological frameworks.

Up to now, a taxonomy of terms to support future research related to digital mobility assessments has still been missing. In our study, we obtained consensus on narrative definitions for the assessment of gait related DMOs acquired in real-world conditions based on an adapted Delphi process. The results of this study have important implications for the development of analysis protocols, as well as for the reporting and comparison of DMOs. Overall, the definitions will allow a more precise use of those terms in future studies, enabling a stronger congruence of clinical, technical, and regulatory activities in this field. Future work within the community includes the refinement of the definitions with respect to concrete study protocols. While supervised gait assessment is currently the reference standard against which future digital mobility measures acquired in the real world will be compared to, validation studies are needed to assess the applicability of the proposed real-world walking definitions. Creating a strong link between supervised and unsupervised gait assessment will ultimately push forward real-world DMO assessment as valid gait and mobility research paradigm.

## Supporting information

**S1 File. Questionnaire with initially proposed definitions of terms related to real-world walking assessed in round one.**
(PDF)

## Acknowledgments

We thank Jeffrey M. Hausdorff (Tel Aviv Sourasky Medical Center, Israel), Inbar Hillel (Tel Aviv Sourasky Medical Center, Israel), Walter Mätzler (University of Kiel, Germany), Kristin Taraldsen (Norwegian University of Science and Technology, Trondheim, Norway), Anisoara Ionescu (Ecole Polytechnique Federale de Lausanne, Switzerland), Kamiar Aminian (Ecole Polytechnique Federale de Lausanne, Switzerland), Brian Caulfield (University College Dublin, Ireland), and Thierry Troosters (Katholieke Universiteit Leuven, Belgium) for their input and comments regarding the initial study design. Full membership of the Mobilise-D consortium is available on the website https://www.mobilise-d.eu/partners. We thank Cameron Kirk for proofreading this manuscript.

## Author Contributions

**Conceptualization:** Felix Kluge, Silvia Del Din, Andrea Cereatti, Heiko Gaßner, Clint Hansen, Jorunn L. Helbostad, Jochen Klucken, Arne Küderle, Arne Müller, Lynn Rochester, Martin Ullrich, Bjoern M. Eskofier, Claudia Mazzà.

**Data curation:** Felix Kluge.

**Formal analysis:** Felix Kluge.

**Investigation:** Felix Kluge.

**Methodology:** Felix Kluge, Silvia Del Din, Jorunn L. Helbostad.

**Supervision:** Claudia Mazzà.

**Writing – original draft:** Felix Kluge, Clint Hansen, Arne Küderle, Martin Ullrich, Claudia Mazzà.

**Writing – review & editing:** Felix Kluge, Silvia Del Din, Andrea Cereatti, Heiko Gaßner, Clint Hansen, Jorunn L. Helbostad, Jochen Klucken, Arne Küderle, Arne Müller, Lynn Rochester, Martin Ullrich, Bjoern M. Eskofier, Claudia Mazzà.

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
