## [Decision Letter · Decision Letter 0]

25 May 2021

Consensus based framework for digital mobility monitoring

PONE-D-21-01788

Dear Dr. Kluge,

We’re pleased to inform you that your manuscript has been judged scientifically suitable for publication and will be formally accepted for publication once it meets all outstanding technical requirements.

Kind regards,

Bruce H. Dobkin, M.D.

Academic Editor

PLOS ONE

2.Thank you for stating the following in the Competing Interests section:

We note that one or more of the authors are employed by a commercial company: Novartis Pharma AG

3. One of the noted authors is a group or consortium [Mobilise-D consortium]. In addition to naming the author group, please list the individual authors and affiliations within this group in the acknowledgments section of your manuscript. Please also indicate clearly a lead author for this group along with a contact email address.

4. Please include additional information regarding the survey or questionnaire used in the study and ensure that you have provided sufficient details that others could replicate the analyses. For instance, if you developed a questionnaire as part of this study and it is not under a copyright more restrictive than CC-BY, please include a copy, in both the original language and English, as Supporting Information.

5. Please provide additional details regarding participant consent. In the ethics statement in the Methods and online submission information, please ensure that you have specified what type you obtained (for instance, written or verbal, and if verbal, how it was documented and witnessed). If your study included minors, state whether you obtained consent from parents or guardians. If the need for consent was waived by the ethics committee, please include this information.

Additional Editor Comments:

thank you for this clarifying expert opinion.
---

## [Author Response · Author response to Decision Letter 0]

3 Jun 2021

Thank you for reviewing and handling our manuscript. We addressed all the remaining requirements and included them into our manuscript. All responses are detailed in the attached file Response_Reviewers.docx.

---

## [Decision Letter · Decision Letter 1]

30 Jun 2021

PONE-D-21-01788R1

Consensus based framework for digital mobility monitoring

PLOS ONE

Dear Dr. Kluge,

Thank you for submitting your manuscript to PLOS ONE. After careful consideration, we feel that it has merit but does not fully meet PLOS ONE’s publication criteria as it currently stands. Therefore, we invite you to submit a revised version of the manuscript that addresses the points raised during the review process.

Judging from the previous decision letter by Dr. Bruce H. Dokin on June 3rd, this Academic Editor, Manabu Sakakibara, found the revised manuscript was sent back without any substantial modification as revision 1. This may be caused by the editorial mistake happened during the Editor exchange. I found both reviewers complained that the revised manuscript was not totally reflected by their critical comments. I have to apology to the authors this inconvenience to take extra time.  

The original two experts in the field have carefully reviewed the manuscript entitled, "Consensus based framework for digital mobility monitoring". Their comments are appended below. 

Both of them acknowledged the manuscript is worth for publication for the field with leaving several serious concerns which should be considered before publication. 

We look forward to receiving your revised manuscript.

Kind regards,

Manabu Sakakibara, Ph.D.

Academic Editor

PLOS ONE

Journal Requirements:

Reviewers' comments:

Reviewer's Responses to Questions

**Comments to the Author**

1. If the authors have adequately addressed your comments raised in a previous round of review and you feel that this manuscript is now acceptable for publication, you may indicate that here to bypass the “Comments to the Author” section, enter your conflict of interest statement in the “Confidential to Editor” section, and submit your "Accept" recommendation.

Reviewer #1: (No Response)

Reviewer #2: (No Response)

2. Is the manuscript technically sound, and do the data support the conclusions?

Reviewer #1: Yes

Reviewer #2: No

3. Has the statistical analysis been performed appropriately and rigorously? 

Reviewer #1: Yes

Reviewer #2: No

4. Have the authors made all data underlying the findings in their manuscript fully available?

Reviewer #1: Yes

Reviewer #2: No

5. Is the manuscript presented in an intelligible fashion and written in standard English?

Reviewer #1: Yes

Reviewer #2: Yes

6. Review Comments to the Author

Reviewer #1: It would appear my original review comments were not included in the decision letter sent to the authors after first review. I have copied these comments again here:

Summary

Authors report the results of a survey on appropriate terminology to use for describing digital mobility assessments for real world walking. Authors use the Delphi process to obtain consensus on terminology from a sample of 162 of their colleagues in the Mobilise-D research project. As acknowledged by the authors the study has limited generalisability as the sample is restricted to one research project and there is a strong risk of ‘group-think’ where students and staff reflect the view of their professors and management. However, the study is a welcome attempt to introduce a taxonomy of terms to support future research around digital mobility assessments. While the focus of the study is on terminology for real world unsupervised gait assessment, this should be placed more clearly in the context of supervised clinical and functional assessment which is the genesis of this work and will remain the reference standard against which unstructured walking will be compared. While the reports of consensus terms are welcome the findings might be more useful is they were accompanied by recommendations on their use in gait assessment studies.

Reviewer #2: There are some weaknesses through the manuscript which need improvement. Therefore, the submitted manuscript cannot be accepted for publication in this form, but it has a chance of acceptance after a minor revision. My comments and suggestions are as follows:

1- Abstract gives information on the main feature of the performed study and it is focused on the background. Therefore, some details about achieved results must be added.

2- Authors must clarify necessity of the performed research. Objectives of the study, and also differences with the previous researches must be clearly mentioned in introduction.

3- The literature study must be enriched. In this respect, authors must read and refer to the following recent and relevant published papers: (a) wearable sensors: https://doi.org/10.1016/j.sna.2020.112105 (b) consensus in probabilistic: https://doi.org/10.1016/j.eswa.2020.114315

4- It is strongly suggest to add figures for better description of concept and some conditions. In addition, statistical analysis is required.

5- In its language layer, the manuscript should be considered for English language editing. There are sentences which have to be rewritten.

6- The conclusion must be more than just a summary of the manuscript. List of references must be updated based on the proposed papers. Please provide all changes by red color in the revised version.

7. PLOS authors have the option to publish the peer review history of their article (what does this mean?). If published, this will include your full peer review and any attached files.

Reviewer #1: No

Reviewer #2: No

---

## [Author Response · Author response to Decision Letter 1]

23 Jul 2021

Dear reviewers,

thank you very much for your valuable comments. We addressed all suggestions as indicated in the attached document.

Best regards, 

Felix Kluge

---

## [Decision Letter · Decision Letter 2]

10 Aug 2021

Consensus based framework for digital mobility monitoring

PONE-D-21-01788R2

Dear Dr. Kluge,

We’re pleased to inform you that your manuscript has been judged scientifically suitable for publication and will be formally accepted for publication once it meets all outstanding technical requirements.

Kind regards,

Manabu Sakakibara, Ph.D.

Academic Editor

PLOS ONE

Additional Editor Comments (optional):

Reviewers' comments:

Reviewer's Responses to Questions

**Comments to the Author**

1. If the authors have adequately addressed your comments raised in a previous round of review and you feel that this manuscript is now acceptable for publication, you may indicate that here to bypass the “Comments to the Author” section, enter your conflict of interest statement in the “Confidential to Editor” section, and submit your "Accept" recommendation.

Reviewer #1: All comments have been addressed

Reviewer #2: All comments have been addressed

2. Is the manuscript technically sound, and do the data support the conclusions?

Reviewer #1: Yes

Reviewer #2: Partly

3. Has the statistical analysis been performed appropriately and rigorously? 

Reviewer #1: Yes

Reviewer #2: I Don't Know

4. Have the authors made all data underlying the findings in their manuscript fully available?

Reviewer #1: Yes

Reviewer #2: Yes

5. Is the manuscript presented in an intelligible fashion and written in standard English?

Reviewer #1: Yes

Reviewer #2: No

6. Review Comments to the Author

Reviewer #1: All comments have been satisfactorily addressed.

No additional comments, best of luck with publication.

Reviewer #2: (No Response)

7. PLOS authors have the option to publish the peer review history of their article (what does this mean?). If published, this will include your full peer review and any attached files.

Reviewer #1: No

Reviewer #2: No

---

## [Editor Report · Acceptance letter]

12 Aug 2021

PONE-D-21-01788R2 

Consensus based framework for digital mobility monitoring 

Dear Dr. Kluge:

I'm pleased to inform you that your manuscript has been deemed suitable for publication in PLOS ONE. Congratulations! Your manuscript is now with our production department. 

Kind regards, 

on behalf of

Dr. Manabu Sakakibara 

Academic Editor

PLOS ONE